# Fishing Innate Immune System Properties through the Transcriptomic Single-Cell Data of *Teleostei*

**DOI:** 10.3390/biology12121516

**Published:** 2023-12-12

**Authors:** Aleksandr V. Bobrovskikh, Ulyana S. Zubairova, Alexey V. Doroshkov

**Affiliations:** 1Department of Physics, Novosibirsk State University, 630090 Novosibirsk, Russia; 2The Federal Research Center Institute of Cytology and Genetics, Siberian Branch of the Russian Academy of Sciences, 630090 Novosibirsk, Russia; ulyanochka@bionet.nsc.ru (U.S.Z.); ad@bionet.nsc.ru (A.V.D.); 3Department of Information Technologies, Novosibirsk State University, 630090 Novosibirsk, Russia; 4Department of Genomics and Bioinformatics, Institute of Fundamental Biology and Biotechnology, Siberian Federal University, 660036 Krasnoyarsk, Russia

**Keywords:** aquaculture, drug development, gene networks, immunity, meta-analysis, model organisms, regeneration, systems biology, teleosts, zebrafish

## Abstract

**Simple Summary:**

The study of the innate immune system’s (IIS) properties is one of the key tasks of modern immunology, since it could help enhance the quality of therapy and drug discovery. This system remains understudied for nonmammalians, and teleost fish are promising model organisms for fundamental studies of IIS organization. The recent single-cell transcriptomics (scRNAseq) experiments carried out on zebrafish and other teleosts brought fundamental knowledge about their innate immunity organization. The aim of this review is to summarize information about the available IIS-related scRNAseq experiments for teleosts and outline further perspectives for studies in this field. We found 89 scRNAseq datasets for zebrafish in different stages and organs suitable for the further meta-analysis of IIS properties and six experiments for other teleosts. Therefore, the obtained data for zebrafish could be further generalized and compared with that of mammals, while for other teleosts, we still have a large gap in knowledge. We believe that in the coming years, new scRNAseq data obtained for nonmodel teleosts will be of particular value and interest in the context of animal immunology.

**Abstract:**

The innate immune system is the first line of defense in multicellular organisms. *Danio rerio* is widely considered a promising model for IIS-related research, with the most amount of scRNAseq data available among *Teleostei*. We summarized the scRNAseq and spatial transcriptomics experiments related to the IIS for zebrafish and other *Teleostei* from the GEO NCBI and the Single-Cell Expression Atlas. We found a considerable number of scRNAseq experiments at different stages of zebrafish development in organs such as the kidney, liver, stomach, heart, and brain. These datasets could be further used to conduct large-scale meta-analyses and to compare the IIS of zebrafish with the mammalian one. However, only a small number of scRNAseq datasets are available for other fish (turbot, salmon, cavefish, and dark sleeper). Since fish biology is very diverse, it would be a major mistake to use zebrafish alone in fish immunology studies. In particular, there is a special need for new scRNAseq experiments involving nonmodel *Teleostei*, e.g., long-lived species, cancer-resistant fish, and various fish ecotypes.

## 1. Introduction

The innate immune system (IIS) is a fundamental way of protecting multicellular organisms from a variety of pathogens and fighting against diseases [1]. This system originated approximately 700–800 million years ago and started with the first phagocytic cells of multicellular organisms [2]. Still, modern studies focus on using mice as a model for human disease, with a particular focus on the adaptive component of the immune system [3,4]. However, the impact of the IIS on certain diseases remains understudied. It is well known that the specific properties of the IIS could make a huge contribution to lifespan and resistance to diseases in mammals, foremost in the case of the naked mole-rat [5]. Spatial and single-cell transcriptomics (scRNAseq) datasets are especially valuable for IIS-related studies, because this system is highly decentralized. ScRNAseq technologies could be effectively used to uncover the properties of populations of teleost immune cells and their interactions during immune response [6].

The zebrafish, *Danio rerio*, is a comprehensively studied model object of fish genetics. The presence of an adaptive component of the immune system in adult zebrafish makes it a promising model for human diseases [7]. At the same time, the larval stages could be used as a model for the isolated IIS response [8]. However, *Danio rerio* remains underutilized in the context of IIS-related studies, e.g., host–microbe interactions [8] and human infections [9]. In recent years, we have seen a growing interest in the evolutionary and ecological aspects of the *Teleostei* immune system due to the growing need to control multiple diseases in aquaculture [10]. In this sense, teleost fish could become a hub taxon in studying the properties of IIS. Presently, more and more details are emerging about the evolution and organization of fish IIS. Obtaining new ScRNAseq data related to the IIS of fish could further extend our knowledge of the general aspects of the organization and functioning of innate immunity.

In this review, we highlight the following strategies for IIS-related scRNAseq studies of *Teleostei*:(I)To perform a meta-analysis using currently available datasets for *Danio rerio*.(II)To obtain new scRNAseq data for nonmodel teleosts.(III)To conduct a comprehensive comparison between zebrafish and mammal IIS, including the evolutionary analysis of IIS-related genes, comparative genomics and transcriptomics, and systems biology approaches.

Therefore, the main focus of our review is to describe and summarize the currently available scRNAseq experiments carried out on *Danio rerio* and other species of *Teleostei* (described in the Section 4 ). Also, we believe that obtaining scRNAseq data on nonmodel teleost fish will provide a systematic view of the regulation and evolution of IIS in this infraclass and will bring immunologists closer to solving the problem of selecting adequate models of innate immunity. In addition, such knowledge will be extremely useful in the aquaculture of economically important fish species. The *Danio rerio* is definitely an excellent model in animal genetics, but it is not fully representative of this taxon, and we cover this aspect in the Section 5.

## 2. A Brief Overview of the *Teleostei* Innate Immune System

*Danio rerio* is the model organism of *Teleostei* used in many innate immunity studies [11]. The main regulatory factors of IIS [12] and immune cell isolation protocols [13] for zebrafish are well-known. There is a high similarity between teleosts and mammals’ complementary systems [14], such as many common pattern recognition receptors [15] and downstream signaling components [16]. Also, homologues of mammalian NOD-like and Toll-like receptors are presented in the fish genome [17], as are many RIG-I-like receptors [16]. Petit et al. [18] identified several candidates for β-glucan receptors in the carp genome and emphasized the general similarity between mammals and fish in CLR-activating pathways. In addition, two homologues of Cyclic GMP-AMP synthase from the cGAS-STING DNA-sensing pathway were identified in zebrafish [19]. Moreover, the major histocompatibility complex (MHC) organization in fish resembles the mammalian one [20]. Murdoch and Rawls emphasized evolutionary conservatism in microbiota perception and response between fish and mammals, especially for microbiota-induced innate immune phenotypes [21]. Summarizing the facts above, there is strong evidence for significant homology in IIS organization between animals and fish.

*Teleostei* lack lymph nodes and bone marrow, but they have kidney marrow as a functional equivalent of human bone marrow [22]. Zebrafish have three main stages of hematopoiesis: **early-embryonic, embryonic, and adult**  [23]. **Early-embryonic** hematopoiesis in zebrafish occurs in two waves: (1) the primitive wave 10–36 h post fertilization (hpf), where the posterior lateral-plate mesoderm (PLPM) forms primitive erythrocytes and neutrophils, and the anterior lateral-plate mesoderm (ALPM) gives rise to primitive macrophages and neutrophils; (2) the definitive wave (23–98 hpf), where PLPM cells pass through the transient stage (24–30) and form erythro-myeloid progenitors, and a subpopulation of cells from the ventral wall of the dorsal aorta (VDA) go through endothelial–hematopoietic transformation and form primitive multipotent hematopoietic stem cells (HSCs) (26–54 hpf), part of which are released and form hematopoietic stem progenitor cells (HSPCs) [23]. **Embryonic** hematopoiesis occurs in caudal hematopoietic tissue (CHT) two days post fertilization (dpf), and through this period, HSCs and HSPCs are able to self-renew and differentiate. **Adult** hematopoiesis is occurring in the thymus at 3 dpf and in the pronephros at 4 dpf. The pronephros accommodates self-renewing and differentiating HSCs and HSPCs and plays the role of the main organ in adult hematopoiesis as mesonephros [23]. Therefore, there are differences in the origin and molecular properties of IIS cells between embryonal, larval, and adult stages of teleosts. Also, the activation of the adaptive component of the immune system of zebrafish occurs in a period 4–6 weeks after fertilization [24].

Teleosts produce all the main types of blood cells of IIS: **macrophages, granulocytes (neutrophils and eosinophils), dendritic cells, B lymphocytes, nonspecific cytotoxic cells, and mast cells** [25]. The key components of the IIS of fish are macrophages and neutrophils [26]. The general, up-to-date overview of the main components of the fish immune system is provided by Mokhtar and coauthors [27].

**Macrophages** are professional phagocytes that play an essential role in the regeneration processes of various tissues and organs (heart, fin, microglia, and others) [28]. Besides their roles in immune response, macrophages connect innate and adaptive components of the teleosts immune system, and their polarization into M1 or M2 types occurs under different stimuli [29]. There is emerging evidence that the metabolic reprogramming of macrophages in teleosts is similar to that in mammals: inflammatory macrophages (M1) are reprogrammed toward glycolysis, and anti-inflammatory macrophages (M2) are reprogrammed toward oxidative phosphorylation [30,31]. Fish macrophages in the liver play a crucial role in the immune response of this organ and could be easily visualized in real time using various fluorescent zebrafish lines, both in the adult and larval stages of development, for modeling various liver pathologies [32].

**Neutrophils** are important players in inflammatory processes against different pathogens in fish. The is clear similarity in the acute inflammatory responses of neutrophils between fish and mammals, but the huge reduction in neutrophil number in the circulating blood of fish compared with mammals was found by Havixbeck and Barreda [33]. Neutrophils are the main controllers of invasive infection and promoters of transformed cell proliferation [26].

**Mast cells** and **eosinophils** in fish are functionally similar to the mast cells of mammals, and an increase in the amount of these cells is usually detected in inflamed tissues [34]. Also, there is evidence of a difference between basophilic and eosinophilic components for various species of fish [34]. The importance of fish mast cells in immune responses and diseases was emphasized in the review by Sfacteria et al. [35]. Specialized defense dendritic cells are Langerhans cells; they recognize foreign antigens in skins and mucous membranes in various organisms, from fish to mammals [36]. These cells are likely to be able to activate T cells by expressing genes related to antigen presentation [37].

Teleost fish have four subpopulations of **B cells**. Three of them exclusively express surface immunoglobulins IgM, IgD, or IgT, and one subpopulation coexpresses surface IgM and IgD [38]. The fundamental mechanisms of immunoglobulin diversity in teleosts are similar to those in mammals [39]. It was found that mammalian B cells are stimulated in mucosa-associated lymphoid tissue (MALT), but in fish, MALT is subdivided into gut-associated, skin-associated, and gill-associated lymphoid tissue [38].

**NK-like cells** are able to kill altered virus-infected and foreign (allogeneic and xenogeneic) target cells [40]. A promising property of NK-like cells is their antitumor activity, which could be precisely studied at the single-cell level in real time [41]. **Nonspecific cytotoxic** (NCC) cells in fish possess killing activity against infected cells, tumors, and parasites [42]. Teng et al. [43] found significant proliferation of NCCs at the early stages of *N. seriolae* infection and CaNCCRP1 as a potential marker of nonspecific cytotoxic cells in fish.

There is limited information about the **eosinophils** and **basophils** of fish, since they are generally considered minor components of IIS. However, a recent study on the teleost fish *Takifugu rubripes* showed the existence of IgM on the surface of basophils. This fact is in favor of the activation of basophils in teleost in an Ab-dependent manner and means there is a possibility of interactions as antigen-presenters between basophils and T-cells [44].

Despite a clear understanding of the overall similarity of the mammal and fish IIS, we lack systematic comparisons between them regarding the cell-type composition of the IIS, the overall plasticity of immune phenotypes, the regulation of immune cells, or the contribution of a particular IIS in certain processes (e.g., inflammation, aging, response to drugs, and cancerogenesis). To explicitly solve this problem, we could use detailed transcriptomic data with single-cell resolution related to the IIS. In recent years, the scientific community has produced many experiments related to the IIS of zebrafish, including scRNAseq datasets, and the main part of our review is focused on describing these data. We also described scRNAseq datasets available for other species of *Teleostei* suitable for IIS-related studies.

## 3. Methodology

We summarized the publicly available transcriptomic datasets with single-cell and spatial resolution for the teleosts related to the IIS cell and stored them in Appendix A. The dataset was considered relevant and appended to the table if the authors of the corresponding papers identified as a cluster at least one of the IIS cell types or T lymphocytes. We also took into consideration scRNAseq experiments, which contain glial cells, as a potential source of residential macrophages.

We searched the appropriate datasets from Gene Expression Omnibus database [45] on 18 September 2023. Our initial query was specific to the following cellular types:
“single cell” (immune OR macrophage OR neutrophil OR eosinophilOR dendritic cells OR lymphocyte OR NK-like cells OR mast cellsOR HSPC OR monocyte OR glia) AND “bony fish” [porgn:__txid7898]AND “Expression profiling by high throughput sequencing” [Filter].

Using the initial query, we found 79 experiments, from which we manually selected 69 suitable ones. To validate the initial request, we used a second query specific to tissues:
“single cell” (kidney OR embryo OR larvae OR intestineOR brain OR spleen OR glia OR head kidney OR liver OR heartOR neuronal retina OR trunk OR gills OR whole body OR thymusOR telencephalon OR blood OR HSPC OR spinal cord OR tail OR thyroid gland)AND “bony fish” [porgn:__txid7898]AND “Expression profiling by high throughput sequencing” [Filter].

The second query found 273 experiments, and after manual selection, 19 of them were added to the final dataset.

Also, we added 7 suitable single-cell experiments for *Danio rerio* from the Single-Cell Expression Atlas [46]. We added, in Appendix A, the following details about the found experiments: identifier, year of publication of data, title of experiment, organism, summary, type of experiment, overall design, contributor(s), citation(s), organ/tissue/cell line, stage, platform of sequencing, and used technologies. Also, at the end of these tables, we included the columns describing the occurrence of the immune cell types mentioned as clusters in the original paper: macrophages, neutrophils, eosinophils, dendritic cells, Langerhans cells, B lymphocytes, NK-like cells, mast cells, and monocytes. We also considered clusters of HSCs/HSPCs cells, T lympocytes, and glial cells in our review, since these cell types could be useful in IIS-related studies. The total number of added experiments in our Appendix A is 95. In total, 89 of them were carried out on zebrafish, among which 58 included the adult stages (described in Appendix A) and 36 included the immature stages of zebrafish (described in Appendix A). Descriptions of the other scRNAseq datasets are given in Appendix A.

## 4. Results

In this section, we summarize the currently available trancriptomic data with single-cell and spatial resolution for the *Teleostei*. Also, we discuss the possible cases for meta-analyses using *Danio rerio* scRNAseq datasets. In parentheses, we provide the statistic of the occurrence of the specific cell type as a cluster in the original papers. For instance, (3/7) means that three of seven experiments in a certain category contain this specific type of immune cell (marked in Appendix A by “+”).

### 4.1. The Available Single-Cell Data for Adult Zebrafish

We found 58 transcriptomic experiments with single-cell or spatial resolution related to the IIS that were conducted on adult zebrafish. A detailed summary of these experiments is given in Appendix A, and a short summary is given in Table 1. The scRNAseq datasets suitable for further meta-analyses (two or more experiments per tissue or organ) are visualized on a colorized dotplot in Figure 1A, and they are the heart, liver, kidney, spleen, intestine, spinal cord, brain, and granuloma. Although this dataset is massive, the distribution of immune cells and tissues in the included experiments is heterogeneous. The majority of studies include IIS cells derived from kidneys (17). Also, there are a large number of scRNAseq datasets from other organs of adult fish: the brain (10), liver (6), heart (6), intestine (5), spinal cord (5), and spleen (3).

Seventeen studies were carried out on the main haematopoietic organ of zebrafish, the **kidney**: GSE100910 [47], GSE100913 [47], GSE112438 [48], GSE130487 [49], GSE150373 [50], GSE151232 [51], GSE166646 [52], GSE176036 [50], GSE179401 [53], GSE183382 [54], GSE190794 [55], GSE191029 [56], GSE200756 [57], E-GEOD-100911 [47], E-MTAB-5530 [58], E-MTAB-7117 [59], E-MTAB-7159 [59]. Most of these studies are concentrated on the **kidney marrow** (10/17) IIS cells. According to the clusters that the authors identified, the majority of IIS cell types were clearly presented: B lymphocytes (12/17), HSPC (12/17), macrophages (11/17), neutrophils (9/17), NK-like cells (8/17), and T lymphocytes (15/17). This subset could be further used for comparison with IIS cells in the bone marrow of mammals based on gene markers and IIS ratios. Hematopoiesis also occurs in the **spleen and thymus** in fish [60]. For the **spleen**, we found three datasets: GSE186158 [61], E-MTAB-4617 [62], and GSE130487 [49]. For the **thymus**, the are two available datasets: E-GEOD-100911 [47] and GSE190794 [55]. In these five experiments, B cells (5/5) and T cells (5/5) were identified, as were other clusters, namely NK-like cells (3/5), HSPCs (2/5), macrophages (2/5), neutrophils (2/5), and dendritic cells (1/5). This subset could be used as a reference in annotations of IIS cell types in other experiments and for extended reconstruction of the developmental trajectories of innate immune cells.

There are numerous datasets of tissues with resident immune cells. ScRNAseq experiments on **intestine** tissue could be used as model tissue of IIS resident cells: GSE130487 [49], GSE135767 [63], GSE165888 [64], GSE184363 [65], and GSE228806 [66]. All the authors of these experiments report clusters of macrophages and B-lymphocytes. Also, clusters of neutrophils (2/5), NK-like cells (2/5), dendritic cells (1/5), HSPCs (1/5), and T lymphocytes (3/5) are presented. ScRNAseq experiments on zebrafish **hearts** could be used to study the role of IIS cells in regeneration processes. We found six experiments, GSE115381 [67], GSE130487 [49], GSE138181 [68], GSE145980 [69], GSE202836 [70], and GSE228806 [66], in which clusters of macrophages (6/6), B lymphocytes (3/6), NK-like cells (2/6), neutrophils (2/6), HSPCs (1/6), dendritic cells (1/6), and T cells (3/6) were described. A recent meta-analysis of heart cells in the regeneration processes of seven experiments revealed the enrichment of processes associated with the immune system, including mast cells [71]. The key advantages of such a meta-analysis include its greater accuracy and sensitivity, as well as the ability to identify internal heterogeneity in data depending on the conditions of the input experiments, which was demonstrated by the study: the authors identified both conservative and injury-specific (resection, genetic ablation, and cryoinjury) genetic systems involved in regenerative processes [71].

The **Liver** of zebrafish is an especially valuable model for drug-induced injuries [72]. We found a subset of six experiments with IIS cells of the liver: GSE130487 [49], GSE150751 [73], GSE181987 [74], GSE192740 [75], GSE217839 [76], GSE228806 [66]. All of these experiments contain clusters of macrophages; other clusters of IIS cells include B cells (4/6), neutrophils (2/6), NK-like cells (2/6), dendritic cells (2/6), HSPCs (1/6), and T cells (4/6).

We also found single datasets including other organs and tissues of fish. HSPC cells were refined in **gills**, GSE198044 [77]; clusters of macrophages and neutrophils were revealed in **thyroid glands**, E-GEOD-133466 [78]. In the ligament of the jaw joint, macrophages, neutrophils, and T cells (GSE224197) were identified [79]. In addition, scRNAseq data from the **whole body** of adult zebrafish are available in experiment GSE102990 [80]. These datasets could be coupled and used to identify rare subtypes of IIS cells. Additionally, these data could help to gain new knowledge about the properties of resident macrophages in different tissues, since macrophages were identified in almost every experiment. We added two scRNAseq experiments from **granuloma tissue**: GSE81913 [81] and GSE161712 [82]. These datasets contain clusters of macrophages and could be further compared with macrophages in normal tissues. Also, there is a distinct dataset of T cells, GSE215189 [83] suitable for refining T cell markers from **lymphoid tissue**.

In addition, we included in our datasets of interest scRNAseq experiments carried out on the **neuronal tissues** of zebrafish. The special group of macrophages is microglia, so these experiments could be used for detailed comparison with kidney-derived macrophages and resident macrophages from other tissues. We found six scRNAseq experiments derived from the **brain** of zebrafish: GSE130487 [49], GSE134706 [84], GSE137525 [85], GSE197673 [86], GSE228806 [66], GSE239410 [87]. This experiment is suitable for the reconstruction of the properties of glial cells (3/6), macrophages (3/6), B lymphocytes (2/6), and T lymphocytes (2/6) and presents clusters of dendritic cells, NK-like cells, and HSPCss. Also, we found other scRNAseq experiments including glial cells from the **telencephalon** (4): GSE161834 [88], GSE179134 [89], GSE212314 [90], GSE225863 [91]; **microglia** (2): GSE120467 [92], GSE164772 [93]; **neuronal retina** (2): GSE202212 [94], GSE226373 [95]; **radial glia** (1): GSE134705 [84]; and **retinal glia**: GSE135406 [96]. Additionally, we found scRNAseq data from the peripheral nervous system. In particular, five datasets were obtained for the **spinal cord**: GSE161642 [97], GSE164944 [98], GSE213435 [99], GSE179096 [100], GSE186163 [100]. Also, one dataset is available for **neural-crest-derived** progenitor cells (E-CURD-123) [101].

To sum up, from teleosts, the greatest variety and number of single-cell IIS-related experiments are available for adult zebrafish. The advantage of these experiments is the large diversity of various cell types from hematopoietic and nonhematopoietic organs, which could be further used in large-scale meta-analyses. In addition, an impressive number of datasets of nervous tissues may be used to study the properties of microglia and compare them with macrophages from other organs.

### 4.2. The Available Single-Cell Data for Immature Zebrafish

We found 36 scRNAseq experiments that included the nonadult stages of zebrafish, such as the embryonic, larval, and juvenile stages. An extended summary is given in Appendix A, and a short summary is given in Table 2. Most of these experiments were carried out on whole fish (14), and a significant number of experiments were found for the central nervous system (8), the blood circulatory system (7), and the caudal fin (3). The statistics of these experiments are visualized in Figure 1B.

Fourteen experiments concentrated on transcriptomic profiling of the **entire body** of embryonic and larval stages of zebrafish: GSE68920 [102], GSE120503 [103], GSE160038 [104], GSE162979 [105], GSE173972 [106], GSE176853 [107], GSE182213 [108], GSE191029 [56], GSE196553 [109], GSE209884 [110], GSE239880 [111], GSE239949 [112], GSE198571 (only larvae) [113], GSE202193 (only embryo) [114]. In 12 of the 14 experiments, the authors identified clusters of macrophages; in half the experiments, neutrophils were identified, and HSPCs were identified in 4 experiments. This subset is suitable for the comparison of IIS cells between resident and nonresident lines. It is worth noting that in these experiments, a much larger proportion of IIS cells are in the progenitor state compared with adult fish. The main challenge in using these experiments for such a meta-analysis is to make a correct classification of IIS cells in different organs and stages of development. To perform this properly, researchers can use annotated clusters from whole-body atlases of mature fish, for instance, GSE102990 [80] or GSE130487 [49], or use the embryo scRNAseq atlas [115].

The properties of the hematopoietic stem and progenitor cells could be extensively studied using the data from seven experiments: GSE92542 (larvae) [116], GSE158099 (embryo, larvae) [117], GSE166900 (larvae) [118], GSE167787 (larvae) [119], GSE186423 (larvae) and GSE186427 (embryo) [120], GSE186565 [121]. Four of the seven experiments contain clusters of neutrophils, and three of them contain macrophage clusters. In this subset, we included experiments from **hemogenic endothelium and dorsal aorta** and experiments with **isolated blood and HSPCs cells**. Macrophages, HSPCs, monocytes, and T lympocytes could be studied using data from three scRNAseq experiments carried out on **caudal fin**: GSE137770 (larvae) [122], GSE146404 (embryo) [123], and GSE158851 (larvae) [124]. Also, we found rare experiments for juvenile **spleen** GSE211396 [125], larval **intestine** GSE150498 [126], and larval **granuloma** GSE81913 [81], which were conducted by only one research group. Additionally, we found an experiment with scRNAseq for the **endothelium of the neural crest**, GSE135246 (larvae) [127], where the authors revealed a specific cluster of neutrophils, which could be compared with the experiments above.

We found eight scRNAseq experiments with fish **neuronal tissues** that were suitable for studying the properties of glial cells in various organs: E-CURD-123 (cranial neural crest, juvenile) [101], GSE105010 (brain, juvenile) [128], GSE132166 (oligodendrocyte precursor cells, larvae) [129], GSE164772 (microglia, juvenile) [93], GSE212314 (telencephalon, juvenile) [90], GSE218107 (neuronal retina, larvae) [130], GSE240026 (spinal cord, embryo) [131], GSE241296 (neurons and glia, larvae) [132].

Thus, the amount of data for immature zebrafish is sufficient to conduct a large meta-analysis of various hematopoietic and nonhematopoietic organs. The main difficulty of carrying out such a meta-analysis is making the precise identification of distinct cell types of IIS in the presence of large amounts of progenitors and their descendants in transition states. Therefore, the expert annotation of cell types in the detected clusters and RNA velocity calculations [133] is especially relevant for such a meta-analysis.

### 4.3. The Available Single-Cell Data for Other *Teleostei*

So far, there is a significant lack of available scRNAseq data for IIS cells derived from nonmodel teleosts. However, we emphasize the high value of such data in the context of IIS evolution and the ecological adaptations of fish. These datasets are detailed in Appendix A.

The IIS cells of cavefish, *Astyanax mexicanus*, derived from the **head kidney** exhibit a more sensitive proinflammatory response to bacterial endotoxins and reduced phagocytosis reactions in comparison with zebrafish (GSE128306) [134]. Also, the authors found that the immune system of cavefish shifted to their adaptive component, mainly to specific T-cell populations. Overall, cavefish could serve as a model for immune system evolution in low parasite abundance [134].

There are two single-cell experiments on IIS cells available for turbot, *Scophthalmus maximus*. In the first one, GSE174019 [135], the authors used fish as a model for dynamic bacterial infection and revealed the bifunctional role of IIS cells related to cytotoxicity and immunoregulation during pathogenesis for subtypes CD8+ CTL, CD4-CD8-T, Th17, and ILC3-2-like. The authors created an immune cell landscape for the **head kidney, spleen, and barrier tissues (posterior intestine, gills)**. In a second study, turbot was used as a trained immunity activation model [136] (GSE195628). The authors revealed the importance of neutrophils as regulators of β-glucan-induced trained immunity and emphasized the critical role of the IL-1R signaling pathway in the induction in trained immunity for bacterial response in teleosts. The authors sequenced the cells from the **head kidney and spleen** and identified clusters of macrophages, neutrophils, dendritic cells, and B and T cells [136].

The sequencing of **head kidney** cells for dark sleeper, *Odontobutis potamophila*, in response to *Aeromonas veronii* [137] (GSE229275) showed a significant increase in neutrophils and a decrease in eosinophils in infected granulocytes, as well as the enhancement of ribosome biogenesis through upregulation of RPS12 and RPL12 and upregulation of the key proinflammatory mediators IL1B and IGHV1-4 and the major histocompatibility class II genes MHC2A and MHC2DAB. The treatment of *Aeromonas veronii* could be used as a model to study mechanisms of such human diseases as bacteremia, gastroenteritis, and soft-tissue infections [138].

Also, different **populations of B cells** were extracted and sequenced from rainbow trout, *Oncorhynchus mykiss*, (GSE158102 [139]). Interestingly, only one marker of B cells in fish, *CXCR4*, corresponds to mammalian B cells. The authors noted that isotype exclusion for the B cells of teleosts is wrong, since they showed clear evidence for rearrangement of V_*L*_J_*L*_C_*L*_ genes (light chains of immunoglobulins).

For *Salmo salar* L., ScRNAseq data were obtained for normal and inoculated specimens with *Aeromonas salaricida* **liver** ( GSE207655 [140]). The authors identified clusters of macrophages, neutrophils, B lymphocytes, and NK-like cells; they also noted the upregulation of specific immune processes in pathogenic conditions, including B cell differentiation and T cell activation.

The diversity of all the main types of IIS cells from **blood** and **head kidney** for *Syngnathus typhle* was described in the scRNAseq analysis by Parker et al. [141]; the bioproject ID is PRJNA781832.

Thus, currently, data for nonmodel teleosts cannot be systematized into any large meta-analysis, since the set of experiments is very modest, especially in comparison with available scRNAseq experiments on zebrafish. Oppositely, the design of the experiments already conducted on zebrafish may lead researchers to conduct comparable experiments on other fish species that differ in IIS characteristics from zebrafish. In the next section, we discuss the importance of further obtaining scRNAseq data for nonmodel teleosts.

## 5. Future Directions

There is a special need for the proper systematization of the obtained scRNAseq data for *Danio rerio*. For instance, the meta-analysis was successfully applied to seven scRNAseq experiments on the regenerating heart of zebrafish [71]. Currently, detailed developmental trajectories could be revealed from scRNAseq data for different hematopoietic organs, namely kidney, spleen, and thymus, which could be followed by the identification of the associated gene networks specific to a particular IIS cell type. Moreover, developmental trajectories could be extensively used for creating novel mathematical models of innate immune cell development and differentiation by taking into account the expression patterns of the main effect genes and regulatory genes [142]. Also, a feasible task at the current time is to identify differences in the properties of macrophages in various tissues, such as the intestine, liver, gills, and fins. To date, an extensive set of toolkits and packages have been developed for conducting this type of analysis [143]. The pipeline for the meta-analysis of scRNAseq was described for a dataset of human kidney cells [144] and could be further adapted for other datasets, including zebrafish. The main steps of the meta-analysis, including batch corrections and clustering, could be obtained with the functions of multiple-dataset integration implemented in the Seurat [145] and Harmony [146] packages. For annotation of IIS cell types, researchers could use specific packages such as singleR [147] and reference datasets for the adult [58] and embryonic stages of zebrafish [115].

Also, we need additional scRNAseq experiments for teleosts in various inflammatory conditions. Sterile inflammation is one of the key drivers of chronic kidney disease and cardiovascular diseases, which is caused by activation of the IIS cells [148]. The induction of trained immunity can cause hyperinflammatory and proatherogenic changes in monocytes and the development of cardiometabolic diseases [149]. Neutrophils and macrophages are central regulators of inflammation associated with nonalcoholic steatohepatitis [150]. Recent studies have discovered a tissue-specific and type-specific immunomodulatory role for individual subtypes of blood vessel endothelial cells [151]. The reconstruction of regulatory gene networks related to specific IIS cell types and immune processes is an effective and descriptive approach that could be used in this area. For instance, the reconstruction of coexpression networks based on scRNAseq data for *Oreochromis niloticus* revealed four hub genes for nonspecific cytotoxic cells [152].

Another fundamental issue that mainly limits our knowledge about the IIS in teleosts is the lack of data for nonmodel species. To date, more than 200 fish genomes have been sequenced [153], but less than 10 species have transcriptomic data with cellular resolution. Obtaining single-cell data for nonmodel teleosts could provide unique opportunities to gain a systematic understanding of how IIS has evolved and developed in different fish taxa, which are contrasting in their characteristics. Also, such data could help establish relationships between the ecological niches of teleosts and the peculiarities of their IIS organization. For example, RIG-I is found in only three orders of teleosts, *Cypriniformes*, *Siluriformes*, and *Salmoniformes* [154], while the total number of teleost orders is about 40 [155]. Also, *Chondrichthyes*, *Actinopterygii*, and *Sarcopterygii* differ in their number of genetic copies of TLR [156]. Due to the significant diversity of fish species, the total number of which exceeds the combined number of mammals, birds, amphibians, and turtles [157], this evolutionary group demonstrates a special variety of both ecological characteristics and genetic peculiarities that could be associated with the special properties of IIS cells. Water is a unique habitat for different morphotypes and ecotypes of fish, varying their body shapes and gills, which results in different locomotion characteristics [158]. The ecotypes of fish greatly affect the number and species composition of their parasites; for example, shallow-spawning fish have more trematodes, while deep-spawning fish have more cestodes [159]. The connectivity of marine populations also affects the levels of their parasitic infection [160]. In addition, the improvement of fish’s parasite avoidance behavior is an alternative evolutionary trade-off strategy to the development of their innate and adaptive immune systems [161].

Various lines of genetically modified zebrafish could serve as adequate models of different human cancers [162]. On the other hand, immunologists can gain another layer of information about cancer genetics in this taxon by producing novel scRNAseq data for nonmodel Teleostei species with contrasting cancer resistance and ecological niches. Long-living fish are characterized by higher numbers of copies of tumor suppressor genes, and vice versa; species with large numbers of oncogenes are characterized by a short lifespan [163]. The average lifespan of zebrafish is about three years, but there are species of teleosts from different orders that significantly exceed this limit, living up to more than 100 years, such as Lake sturgeon (*Acipenser fulvescens*), White sturgeon (*Acipenser transmontanus*), Beluga sturgeon (*Huso huso*), Rougheye rockfish (*Sebastes aleutianus*), Pacific Ocean Perch (*Sebastes alutus*), Rebanded Rockfish (*Sebastes babcocki*), Yelloweye Rockfish (*Sebastes ruberrimus*), Shortraker Rockfish (*Sebastes borealis*), Sablefish (*Anoplopoma fimbria*), Orange Roughy (*Hoplostethus atlanticus*), Warty Oreo (*Allocyttus verrucosus*), and Black Cardinal fish (*Epigonus telescopus*) [164]. Also, long-living teleosts could become unique model organisms for studying the senescence processes of immunity using scRNAseq technologies; for example, *Ictiobus cyprinellus* does not demonstrate age-related declines in immune functions [165]. The main markers of immunoaging are a decrease in the number of T cells and a chronic low-grade inflammatory state, which leads to greater cancer susceptibility and infection in older adults [166]. Single-cell data for IIS in humans and mice have contributed much new information to our understanding of the effects of aging on IIS cells and have allowed us to summarize quantitative and molecular changes in major immune cell populations [167]. Thus, the main features of long-living fish resemble another model animal, the naked mole-rat, and the amazing longevity of this rodent is associated with the unique properties of its innate immunity [5,168,169]. However, the widespread usage of the naked mole-rat as a model organism is complicated by several reasons: (i) the long breeding period, because there is only one breeding female per colony and a long gestation period; (ii) the need for a special living place in the form of a large system of cells interconnected by tunnels and the need for climate control [170], (iii) difficulties with transgenesis, and (iv) the high vulnerability of this species to viruses [171]. Therefore, with the recent growth of aquaculture for long-living teleost species, such as *Huso huso* [172], they can become an alternative model organism for studying the aging processes and properties of the immune system. Also, fish species with contrasting lifespans, ecological niches, and other characteristics may differ in the ratios of different types of IIS cells and in their transcriptomic and metabolomic characteristics. In addition, it is known that chronic stress in both mammals and teleosts causes immunosuppression, but the mechanisms of acute response in teleosts have not yet been revealed [173]. Therefore, obtaining new single-cell atlases in fish under different stress conditions could serve as a model that allows for the estimation of the contribution of IIS cell types to stress response and adaptation processes, as well as a deeper understanding of the effect of acute stress on teleosts. Eventually, with comprehensive data gained only on zebrafish, current fish immunology lacks the fundamental knowledge of the innate immunity of other teleosts. For further breakthroughs in this area and the development of new human-like models of various diseases, it is necessary to conduct a series of experiments with single-cell sequencing on nonmodel teleosts that contrast in characteristics from zebrafish. Such further studies with single-cell resolution for IIS for other teleosts could expand our understanding of the processes of pathogenesis, cancerogenesis, aging, and the regeneration of individual tissues, as well as simplify the drug testing process.

## 6. Conclusions

The innate immune system is one of the oldest ways of fighting against various pathogens in multicellular organisms. The evolutionary conservatism of the main pathways of IIS could serve in favor of using many nonmammal species as models for multiple diseases and to develop new drugs. Zebrafish are considered very promising for this purpose, having a short life cycle, broad regeneration ability, multiple options for genetic modifications, and a lack of adaptive immunity in the larval stage [7,8]. While the IIS of zebrafish has been narrowly studied using scRNAseq and spatial transcriptomics, we still have a significant lack of such data for other teleosts. The currently available scRNAseq datasets for *Danio rerio* are sufficient for meta-analysis and evolutionary comparisons. Also, these experiments could further serve to systematize the properties of the IIS of teleosts and could be taken into account during the design of novel studies for other *Teleostei*.

Special attention should be paid to obtaining novel data for contrasting fish species. Teleosts demonstrate wide variability in their immune system and response to pathogens, lifespan, and cancer susceptibility, which makes them a promising object for studying the processes of senescence, regeneration, and resistance to multiple diseases in the context of the IIS. Such data could provide valuable knowledge about the functions of the IIS and establish the relationship between the evolutionary features of fish and the characteristics of their immune system. In addition to its fundamental component, such research can bring positive economic benefits, allowing the introduction of new strategies for controlling parasites in aquaculture farms [174].

## Figures and Tables

**Figure 1 biology-12-01516-f001:**
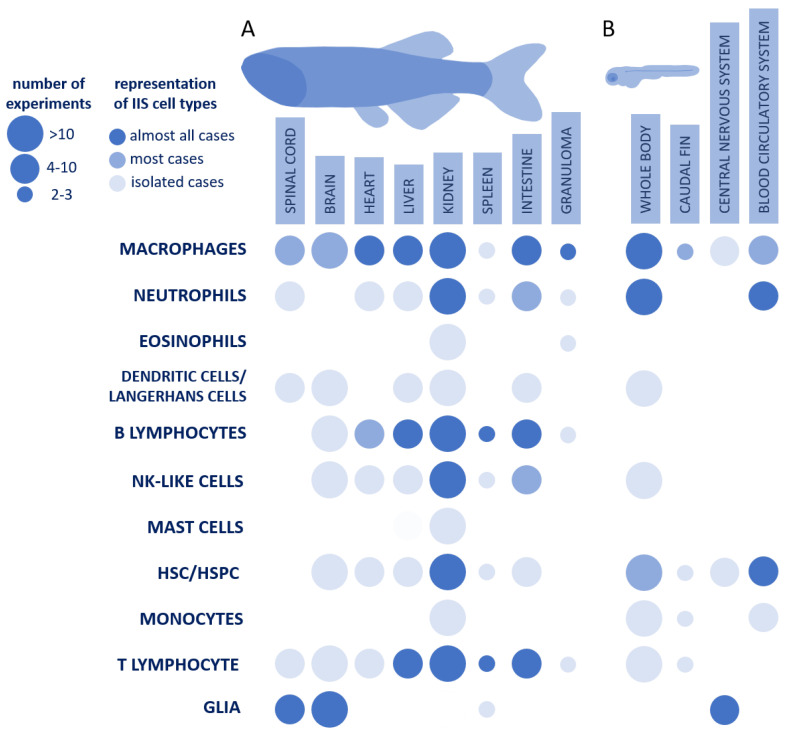
Availability of spatial/scRNAseq data of IIS cells for adult (**A**) and nonmature zebrafish (**B**) in different organs, which could be used for further meta-analyses. The data were collected on 18 September 2023.

**Table 1 biology-12-01516-t001:** IIS-related scRNAseq and spatial transcriptomics experiments on adult zebrafish.

Tissues	Identifiers of Experiments and Their Links
Kidney (17)	E-GEOD-100911, E-MTAB-5530, E-MTAB-7117, E-MTAB-7159, GSE100910, GSE100913, GSE112438, GSE130487, GSE150373, GSE151232, GSE166646, GSE176036, GSE179401, GSE183382, GSE190794, GSE191029, GSE200756
Brain (10)	GSE130487, GSE134706, GSE137525, GSE161834, GSE179134, GSE197673, GSE212314, GSE225863, GSE228806, GSE239410
Heart (6)	GSE115381, GSE130487, GSE138181, GSE145980, GSE202836, GSE228806
Liver (6)	GSE130487, GSE150751, GSE181987, GSE192740, GSE217839, GSE228806
Intestine (5)	GSE130487, GSE135767, GSE165888, GSE184363, GSE228806
Spinal cord (5)	GSE161642, GSE164944, GSE179096, GSE186163, GSE213435
Spleen (3)	E-MTAB-4617, GSE130487, GSE186158
Microglia (2)	GSE120467, GSE164772
Granuloma (2)	GSE81913, GSE161712
Neuronal retina (2)	GSE202212, GSE226373
Gills (1)	GSE198044
Whole body (1)	GSE102990
Thyroid gland (1)	E-GEOD-133466
Jaw joint (1)	GSE224197
Retinal glia (1)	GSE135406
Radial glia (1)	GSE134705
Cranial neural crest (1)	E-CURD-123
Lymphoid tissues (1)	GSE215189

**Table 2 biology-12-01516-t002:** IIS-related scRNAseq and spatial transcriptomics experiments on immature zebrafish.

Tissues	Identifiers of Experiments and Their Links
Whole body (14)	GSE68920, GSE120503, GSE160038, GSE162979, GSE173972, GSE176853, GSE182213, GSE191029, GSE196553, GSE198571, GSE202193, GSE209884, GSE239880, GSE239949
Central nervous system (8)	E-CURD-123, GSE105010, GSE132166, GSE164772, GSE212314, GSE218107, GSE240026, GSE241296
Blood circulatory system (7)	GSE92542, GSE158099, GSE166900, GSE167787, GSE186423, GSE186427, GSE186565
Caudal fin (3)	GSE137770, GSE146404, GSE158851
Granuloma (1)	GSE81913
Endothelium of the neural crest (1)	GSE135246
Intestine (1)	GSE150498
Spleen (1)	GSE211396

## Data Availability

Data available in a publicly accessible repository: Gene Expression Omnibus database at [45] and Single-Cell Expression Atlas at [46]. The identifiers of particular experiments and corresponding articles discussed in this review are available in Appendix A.

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
