# Peer review of "Fishing Innate Immune System Properties through the Transcriptomic Single-Cell Data of Teleostei"

_biology, 2023, doi:10.3390/biology12121516_

Round 1
Reviewer 1 Report
Comments and Suggestions for Authors
1. In general view, the manuscript should be reformulated according to the author's guideline of the Biology journal for the article review type, consisting of at least the Summary, Abstract, Introduction, Content of review results and discussions, and Conclusions, see the previously published article (review type) in MDPI Biology Journal
2. Title
It needs to be revised to make it more focused by adding e.g. 'using Zebrafish (Danio rerio) as a model?', so it could be
'Fishing the innate immune system properties through transcriptomic single cell data using Zebrafish (Danio rerio) as a model' ?
Abstract:
2. The abstract should be improved. It needs to be reformulated with a short background, methods, results and discussion, and conclusions.
3. Keywords: please check the number of keywords for the abstract
Contents:
4. Please double-check the scientific names of species discussed in this manuscript, e.g. in L372 Nile tilapia (?) is not a species name for a fish
5. Conclusions
- please add a conclusion for this manuscript
Author Response
Dear reviewer,
We thank you sincerely for your valuable comments about our work.
We have taken them into account and improved our paper.
Below we provided the answers (A) to your questions (Q).
Q1. In general view, the manuscript should be reformulated according to the author's guideline of the Biology journal for the article review type, consisting of at least the Summary, Abstract, Introduction, Content of review results and discussions, and Conclusions, see the previously published article (review type) in MDPI Biology Journal
A1. Thank you. We extensively reformulated our paper according to previously published reviews in MDPI. Now, our manuscript includes sections: Simple Summary, Abstract, Introduction, The brief overview of Teleostei innate immune system, Methodology, Results, Future Directions and Conclusions. We are sure that the new structure of our review will be much cleaner for readers.
Q2. Title. It needs to be revised to make it more focused by adding e.g. 'using Zebrafish (Danio rerio) as a model?', so it could be 'Fishing the innate immune system properties through transcriptomic single cell data using Zebrafish (Danio rerio) as a model' ?
A2. Thanks for this idea. However, we want to keep the original title of the publication because of the general message we give to the reader in Future Directions and Conclusions is that there is an urgent need to obtain data for non-model types of teleosts. “Another fundamental issue which mainly limits our knowledge about properties of IIS of teleosts is lack of data for non-model species”, “...immunologists can gain another layer of information about the cancer genetics by conducting comprehensive analysis of fish species resistant to cancer and comparing transcriptomes of their IIS cells with IIS cells of species prone to cancer”, “the main features of long-lived fish resemble another model animal, naked mole-rat, and the amazing longevity of this rodent is associated with the unique properties of its innate immunity ”, “The special attention should be paid to get novel data for contrasting fish species. Teleosts demonstrate wide variability in their immune system and response to pathogens, lifespan, and cancer susceptibility, which makes them a promising object for studying processes of senescence, regeneration, and resistance to multiple diseases in the context of IIS.”
Q3. Abstract. The abstract should be improved. It needs to be reformulated with a short background, methods, results and discussion, and conclusions.
A3. Thank you. We improved our abstract, now it contains short background, methods, results and discussion, conclusions. The new abstract is provided below:
“The innate immune system (IIS) is a first line of defense in multicellular organisms. For a long time, Danio rerio has been considered as a promising model for IIS-related studies. To date, scRNAseq experiments of IIS cells have been carried out on zebrafish and non-model teleosts. In this review we collected and summarized information about available transcriptomic experiments forTeleostei (primarily scRNAseq) from the GEO NCBI and the Single Cell Expression Atlas. Most of the experiments we found were carried out at different stages of zebrafish development, in organs such as the kidney, liver, stomach, heart, brain, and could be further used in a large-scale meta-analysis. Only a small number of scRNAseq are available for other fishes (turbot, salmon, cavefish, dark sleeper). However, fish biology is very diverse, and it would be a major mistake to focus only on the zebrafish as a model of IIS. Therefore, there is a special need to obtain new scRNAseq information on non-model Teleostei, including long-lived species, cancer-resistant fishes and various ecotypes. In addition, a meta-analysis of the currently available scRNAseq datasets on zebrafish will clarify the fundamental characteristics of the IIS and assess the conservatism of the system compared to mammals.”
Q4. Keywords: please check the number of keywords for the abstract
A4. Thanks, we checked the number of keywords - we used the 10 keywords and did not exceed the limitation (10).
Q5. Contents. Please double-check the scientific names of species discussed in this manuscript, e.g. in L372 Nile tilapia (?) is not a species name for a fish
A5. Thanks, we checked our manuscript extensively and corrected such errors.
Q6. Conclusions. please add a conclusion for this manuscript
A6. Thank you! We added conclusion to our manuscript:
The innate immune system is one of the oldest ways to fight against various pathogens in multicellular organisms. The evolutionary conservatism of the main pathways of IIS allows the use of many non-mammal species as models for multiple diseases and to develop new drugs. Zebrafish has been considered a very promising model for this purpose, having a short life cycle, broad regeneration ability, multiple options for genetic modifications, and a lack of adaptive immunity in the larval stage [7,8 ]. While the IIS of zebrafish has been narrowly studied using scRNAseq and spatial transcriptomics, we still have a significant lack of such data for other teleosts. The currently available scRNAseq datasets for Danio rerio are sufficient for meta-analysis and evolutionary comparisons. These experiments could further serve as a base for systematizing the properties of IIS of teleosts and could be taken into account during the design of novel studies for other Teleostei.
Special attention should be paid to getting novel data for contrasting fish species. Teleosts demonstrate wide variability in their immune system and response to pathogens, lifespan, and cancer susceptibility, which makes them a promising object for studying processes of senescence, regeneration, and resistance to multiple diseases in the context of IIS. Such data could provide valuable knowledge about the functions of IIS and establish the relationship between the evolutionary features of fish and the characteristics of their immune system. In addition to the fundamental component, such research can bring positive economic benefits, allowing the introduction of new strategies for controlling parasites in aquaculture farms [173].
Reviewer 2 Report
Comments and Suggestions for Authors
The manuscript summary the application of transcriptomic single cell data for uncovering the innate immune system of fish, which is useful to expand our understanding of fish immunity. But most description just focus on the zebrafish, the transcriptomic single cell data of other fish should be mentioned in this review. Also, the information regarding innate immune signal pathway should be added.
1. Using single cell transcriptomic analysis, most cell types have been identified in fish, how about the difference of the innate immune cells population between fish and human? 2. As the important components of innate immunity, more information regarding the cell type of macrophages or NK-like (NCC) should be mentioned; 3. The figures or tables conclude the information described above should be added. Comments on the Quality of English LanguageMinor editing of English language required.
Author Response
Dear reviewer,
We thank you sincerely for your valuable comments about our work.
We have taken them into account and improved our paper..
Below we provided the answers (A) to your questions (Q).
Q1:. Using single cell transcriptomic analysis, most cell types have been identified in fish, how about the difference of the innate immune cells population between fish and human?
A1: Thank you so much for this interesting question. We can not directly answer, but we think that the meta-analysis of available scRNAseq of zebrafish definetely will help to know this. This question leads us to make the following conclusion in the Brief Overview section:
“Despite a clear understanding of the overall similarity of the mammal and fish IIS, we lack systematic comparisons between them regarding the cell-type composition of the IIS, the overall plasticity of immune phenotypes, the regulation of immune cells, or the contribution of particular IIS in certain processes (e.g., inflammation, aging, response to drugs, cancerogenesis). To explicitly solve this problem, we could use detailed transcriptomic data with single-cell resolution related to IIS. In recent years, the scientific community has produced many experiments related to the IIS of zebrafish, including scRNAseq datasets, and the main part of our review is focused on describing these data. We also described scRNAseq datasets available for other species of \textit{Teleostei} suitable for IIS-related studies”
Also, we formulated the following paragraph in Introduction with regards to your question.
The zebrafish, Danio rerio, is a comprehensively studied model object of fish genetics. The presence of an adaptive component of the immune system in adult zebrafish makes it a promising model for human diseases [7]. At the same time, the larval stages could be used as a model for the isolated IIS response [8]. However, Danio rerio remains underutilized in the context of IIS-related studies, e.g., host-microbe interactions [8] and human infections [9]. In recent years, we have seen a growing interest in the evolutionary and ecological aspects of the Teleostei immune system due to the growing need to control multiple diseases in aquaculture [10]. In this sense, teleost fish could become a hub taxon in studying the properties of IIS. To the present day, more and more details are emerging about the evolution and organization of fish IIS. Obtaining new ScRNAseq data related to the IIS of fish could further extend our knowledge about the general aspects of the organization and functioning of innate immunity.
Q2. As the important components of innate immunity, more information regarding the cell type of macrophages or NK-like(NCC) should be mentioned;
A2. Thank you! We added this information in following sentences:
There is emerging evidence that metabolic reprogramming of macrophages in teleosts is similar to that in mammals: inflammatory macrophages (M1) are reprogrammed toward glycolysis, and anti-inflammatory macrophages (M2) are reprogrammed toward oxidative phosphorylation [30, 31].
A promising property of NK-like cells is their antitumor activity, which could be precisely studied at the single cell level in real time [41].
Non-specific cytotoxic (NCC) cells in fish possess killing activity against infected cells, tumors, and parasites [42].
Q3. The figures or tables conclude the information described above should be added.
A3. Thank you! We added Table 1 which summarized available scRNAseq/spatial experiments of IIS-related studies for adult zebrafish in regards to organs as well as Table 2 for immature zebrafish experiments with links to the experiments in databases.
Reviewer 3 Report
Comments and Suggestions for Authors
The aim of the present review is to analyse the RNAseq experiments carried out for zebrafish and other teleost species focused on innate immune system. The article is nicely presented and well-planned. However, I have several concerns indicated below:
· The title of the first paragraph is incorrect because it does not describe the immune system components of zebrafish but of teleost fishes in general, with some specific references to zebrafish. I suggest correcting the title of even deleting itself;
· The results from some references must be described in a more accurate way so that the reader can clearly understand the importance of the cited study in support of the specific section of the article. Two major examples are reference 21 cited in lines 69-71 and reference 42 citated in lines 121-123. The description is not scientifically sound and must be differently stated.
· The authors make a list of the RNAseq experiments relative to each section of the article (for example line 180-183). Although they have attached very detailed tables in the supplementary materials, it would be useful for the reader to have a summary table of the experiments directly in the article, for a more immediate understanding of the numerous information reported.
My raccomandation is to ask the authors for major revisions.
Comments on the Quality of English LanguageI suggest the text should also undergo moderate English revisions.
Author Response
Dear reviewer,
We thank you sincerely for your valuable comments about our work.
We have taken them into account and improved our paper.
Below we provided the answers (A) to your questions (Q).
Q1. The title of the first paragraph is incorrect because it does not describe the immune system components of zebrafish but of teleost fishes in general, with some specific references to zebrafish. I suggest correcting the title of even deleting itself;
A1. Thank you! We changed this title to “The brief overview of Teleostei innate immune system” and separated the Introduction section where we discussed zebrafish in the context of IIS.
Q2. The results from some references must be described in a more accurate way so that the reader can clearly understand the importance of the cited study in support of the specific section of the article. Two major examples are reference 21 cited in lines 69-71 and reference 42 cited in lines 121-123. The description is not scientifically sound and must be differently stated.
A2. Thank you so much! We corrected these sentences and made the extensive overall text improvements. First sentence was rewritten: “Fish macrophages in the liver play a crucial role in the immune response of this organ and could be easily visualized in real time using various fluorescent zebrafish lines, both in adult and larval stages of development, for modeling various liver pathologies [32].” Second one also was rewritten: “Murdoch and Rawls [21] emphasized evolutionary conservatism in microbiota perception and response between fish and mammals, especially microbiota-induced innate immune phenotypes.”
Q3. The authors make a list of the RNAseq experiments relative to each section of the article (for example line 180-183). Although they have attached very detailed tables in the supplementary materials, it would be useful for the reader to have a summary table of the experiments directly in the article, for a more immediate understanding of the numerous information reported.
A3. Thank you! We added Table 1 which summarized available scRNAseq/spatial experiments of IIS-related studies for adult zebrafish in regards to organs as well as Table 2 for immature zebrafish experiments with links to the experiments in databases.
Reviewer 4 Report
Comments and Suggestions for Authors
Comments:
The manuscript biology-2687880 provides information about “Fishing the innate immune system properties through transcriptomic single cell data”. It is an exciting ms, and it is well written. However, the current form of ms is not suitable for publication yet as some minor issues are found in the text. The ms should be improved by addressing those issues before being accepted for publication. In addition, several grammatical errors are found, so please review the manuscript and correct them.
Abstract
Ln 6 – “as very promising” should be “a very promising”
Ln 7 – “as lacking of adaptive immunity” should be “ a lack of adaptive immunity”
Ln 8 – “on larval stage” should be “in the larval stage”
Ln 9- “susceptibility to cancer” should be “cancer susceptibility”
Ln 11 – “for precised studies” should be “for precise studies”
Lns 12-16 - Rephrase the sentences!
Keywords
Lns 17-18 - to be arranged in alphabetical order and make sure they will be different from those in the paper’s title
Ln 21 – “fight against diseases” should be “fighting against diseases”
Ln 22 – “700-800 millions years ago started” should be “700-800 million years ago and started”
Ln 25- “in certain diseases” should be “on certain diseases”
Ln 33 – “specie” should be “species”
Ln 37- “of human” should be “for human”
Ln 41- “lacking” should be “lack”
Ln 41 – “having” should be “have”
Ln 43- “ haematopoiesis” should be “hematopoiesis”. Apply this for entire text!
Ln 47- “pass” should be “passing”
Ln 49 – “endothelial-haematopoietic” should be “endothelial-hematopoietic”
Ln 54- “Adult haematopoiesis is occuring” should be “Adult hematopoiesis is occurring”
Lns 60-69 - modify the sentences!
Ln 71 – “model of” should be “model for”
Lns 74-92- rephrase the sentences to make it clear!
· The first noun, modifier, verb, or pronoun must agree in number with the others in a clause.
· An uncountable noun refers to the whole of something that cannot be separated into parts
· Articles, like a, an, and the, show noun specificity.
· In a list of three or more items, use a comma before the conjunction separating the final item
Lns 95-98 - Rephrase the sentences!
Use prepositions only when they are needed to show relationships between nouns and other words in the sentence
Lns 106-110- rewrite the sentence to reduce complexity!
Ln 112- “receptors of mammals” should be “receptors in mammals”
Ln 114- “seems” should be “seem”
Ln 115-119- modify the sentences!
Ln 125 –“ resembles mammals” should be –“ resemble mammals”
Lns 127 – “lacking” should be “a lack”
Ln 140- “teleosts” should be “teleost”
Ln 141 – “In this paragraph we summarize” should be “In this paragraph, we summarize”
Ln 142 – “of the single cell” should be “on the single cell”
Lns 147-148 - Rephrase the sentence to reduce complexity!
Ln 169- “ includes” should be “including “
Lns 265-266 - rewrite the sentence!
Ln 269- “of of individual” should be “of individual”
Ln 275- “macrophages clusters” should be “macrophage clusters”
Ln 277- “data of three” should be “data from three”
Lns 280-281 - modify the sentence!
Ln 285- “suitable to study” should be “suitable for study”
Ln 293 “to precised” should be “to precise”
Lns 299-301- Rephrase the sentence!
Lns 356-360- modify the sentences!
Ln 368- “was reviewed” should be “were reviewed”
Lns 372-373 - rewrite the sentence!
Lns 380 – 382- Rephrase the sentence!
Ln 421 – “specie to viruses” should be “species to viruses”
Ln 423 – “for study” should be “for studying “
Ln 424- “lifespan” should be “lifespans”

Comments on the Quality of English LanguageExtensive editing of English language required
Author Response
Dear reviewer,
We thank you sincerely for your valuable comments about our work.
We have taken them into account and improved our manuscript. Our detailed answers are marked by (A)
Abstract
Ln 6 – “as very promising” should be “a very promising” - done
Ln 7 – “as lacking of adaptive immunity” should be “ a lack of adaptive immunity” - done
Ln 8 – “on larval stage” should be “in the larval stage” - done
Ln 9- “susceptibility to cancer” should be “cancer susceptibility” - done
Ln 11 – “for precised studies” should be “for precise studies” - done
Lns 12-16 - Rephrase the sentences! - done
(A1) Thanks, we extensively improved our abstract and these errors are not presented in the new version of our manuscript. The new version of abstract is below:
- “The innate immune system (IIS) is a first line of defense in multicellular organisms. For a long time, Danio rerio has been considered as a promising model for IIS-related studies. To date, scRNAseq experiments of IIS cells have been carried out on zebrafish and non-model teleosts. In this review we collected and summarized information about available transcriptomic experiments forTeleostei (primarily scRNAseq) from the GEO NCBI and the Single Cell Expression Atlas. Most of the experiments we found were carried out at different stages of zebrafish development, in organs such as the kidney, liver, stomach, heart, brain, and could be further used in a large-scale meta-analysis. Only a small number of scRNAseq are available for other fishes (turbot, salmon, cavefish, dark sleeper). However, fish biology is very diverse, and it would be a major mistake to focus only on the zebrafish as a model of IIS. Therefore, there is a special need to obtain new scRNAseq information on non-model Teleostei, including long-lived species, cancer-resistant fishes and various ecotypes. In addition, a meta-analysis of the currently available scRNAseq datasets on zebrafish will clarify the fundamental characteristics of the IIS and assess the conservatism of the system compared to mammals.”
Keywords
Lns 17-18 - to be arranged in alphabetical order and make sure they will be different from those in the paper’s title
(A2) Thanks, we checked our keywords and placed them in alphabetical order
Ln 21 – “fight against diseases” should be “fighting against diseases” - done
Ln 22 – “700-800 millions years ago started” should be “700-800 million years ago and started” - done
Ln 25- “in certain diseases” should be “on certain diseases” - done
Ln 33 – “specie” should be “species” - done
Ln 37- “of human” should be “for human” - done
Ln 41- “lacking” should be “lack” - done
Ln 41 – “having” should be “have” - done
Ln 43- “ haematopoiesis” should be “hematopoiesis”. Apply this for entire text! - done
Ln 47- “pass” should be “passing” - done
Ln 49 – “endothelial-haematopoietic” should be “endothelial-hematopoietic” - done
Ln 54- “Adult haematopoiesis is occuring” should be “Adult hematopoiesis is occurring” - done
(A3) Thanks, we corrected these errors including the errors presented in other places in our text.
Lns 60-69 - modify the sentences!
(A4) Thanks, we modified these sentences:
“Teleosts produce all the main types of blood cells of IIS: macrophages, granulocytes (neutrophils and eosinophils), dendritic cells, B lymphocytes, non-specific cytotoxic cells, and mast cells [25]. The key components of the IIS of fish are macrophages and neutrophils [26]. The general, up-to-date overview of the main components of the fish immune system is provided by Mokhtar and co-authors [27].
Macrophages are professional phagocytes that play an essential role in the regeneration processes of various tissues and organs (heart, fin, microglia, and others) [28]. Besides their roles in immune response, macrophages connect innate and adaptive components of the teleosts immune system, and their polarization into M1 or M2 types occurs un der different stimuli [29]. There is emerging evidence that metabolic reprogramming of macrophages in teleosts is similar to that in mammals: inflammatory macrophages (M1) are reprogrammed toward glycolysis, and anti-inflammatory macrophages (M2) are reprogrammed toward oxidative phosphorylation [30,31]. Fish macrophages in the liver play a crucial role in the immune response of this organ and could be easily visualized in real time using various fluorescent zebrafish lines, both in adult and larval stages of development, for modeling various liver pathologues [32].
Ln 71 – “model of” should be “model for” - corrected
Lns 74-92- rephrase the sentences to make it clear!
(A5) Thanks, we rephrased these sentences:
Neutrophils are important players in inflammatory processes against different pathogens in fish. The clear similarity in acute inflammatory responses of neutrophils between fish and mammals but the huge reduction in the neutrophil number in the circulating blood of fish compared to mammals were found by Havixbeck and Barreda [33]. Neutrophils are the main controllers of invasive infection and promoters of transformed cell proliferation [26].
Mast cells and eosinophils in fish are functionally similar to the mast cells of mammals, and an increase in amounts of these cells is usually detected in inflamed tissues [34]. Also, there is evidence of a difference between basophilic and eosinophilic components for various species of fish [34]. The importance of fish mast cells in immune responses and diseases was emphasized in the review by Sfacteria et al. [35]. Specialized defense dendritic cells are Langerhans cells; they recognize foreign antigens in skins and mucous membranes in various organisms, from fish to mammals [36]. These cells are likely able to activate T cells by expressing genes related to antigen presentation [37].
Teleost fish have four subpopulations of B cells. Three of them exclusively express surface immunoglobulins IgM, IgD, or IgT, and one subpopulation coexpresses surface IgM and IgD [38]. The fundamental mechanisms of immunoglobulin diversity in teleosts are similar to those in mammals [39]. It was found that mammalian B cells are stimulated in mucosa-associated lymphoid tissue (MALT), but in fish, MALT is subdivided into gut- associated, skin-associated, and gill-associated lymphoid tissue [38].
NK-like cells are able to kill altered virus-infected and foreign (allogeneic and xeno- geneic) target cells [40]. A promising property of NK-like cells is their antitumor activity, which could be precisely studied at the single cell level in real time [41]. Non-specific cytotoxic (NCC) cells in fish possess killing activity against infected cells, tumors, and parasites [42].
Lns 95-98 - Rephrase the sentences!
(A6) Thanks, we rephrased these sentences:
There is limited information about eosinophils and basophils of fish since they are generally considered minor components of IIS. However, the recent study on the teleost fish Takifugu rubripes showed the existence of IgM on the surface of basophils. This fact indicates in favor of activation of basophils in teleost in an Ab-dependent manner and means the possibility of interactions as antigen-presenters between basophils and T-cells [44].
Lns 106-110- rewrite the sentence to reduce complexity!
(A7) Thanks, we reduced the complexity of this sentence:
“There is a high degree of similarity between teleosts and mammals complement systems [14], many common pattern recognition receptors [15] as well as downstream signaling components between zebrafish and mammals [16]. Also, homologues of mammalian NOD-like and Toll-like receptors are presented in the fish genome [17], as are many RIG-I-like receptors [16].“
Ln 112- “receptors of mammals” should be “receptors in mammals” -done
Ln 114- “seems” should be “seem” -done
Ln 115-119- modify the sentences!
(A8) Thanks, we modified these sentences:
Dectin-1 is a well-known member of the family of C-type lectin receptors (CLRs) in mammals that is still not found in fish. However, Petit et al. [18] identified several candidate β-glucan receptors in the carp genome and emphasized the general similarity between mammals and fish in CLR activating pathways.
Ln 125 –“ resembles mammals” should be –“ resemble mammals” - done
Lns 127 – “lacking” should be “a lack” - done
Ln 140- “teleosts” should be “teleost” - done
Ln 141 – “In this paragraph we summarize” should be “In this paragraph, we summarize” - done
Ln 142 – “of the single cell” should be “on the single cell” -done
Lns 147-148 - Rephrase the sentence to reduce complexity!
(A9) Thanks, we rephrased this sentence:
“The dataset was considered relevant and appended to the table if the authors of the corresponding papers identified as a cluster at least one of the IIS cell types or T lymphocytes. We also take into consideration scRNAseq experiments, which contain glial cells, as a potential source of residential macrophages.”
Ln 169- “ includes” should be “including “ - done
Lns 265-266 - rewrite the sentence!
(A10) Thanks, we rewrited this sentence:
“It is worth noting that in these experiments, a much larger proportion of IIS cells are in the progenitor state compared to adult fish.”
Ln 269- “of of individual” should be “of individual” - done
Ln 275- “macrophages clusters” should be “macrophage clusters” - done
Ln 277- “data of three” should be “data from three” - done
Lns 280-281 - modify the sentence!
(A10) Thanks, we modified this sentence:
“Macrophages, HSPCs, monocytes, and T lympocytes could be studied using data from three scRNAseq experiments carried out on caudal fin: GSE137770 (larvae) [122], GSE146404 (embryo) [123], and GSE158851 (larvae) [124].”
Ln 285- “suitable to study” should be “suitable for study” - done
Ln 293 “to precised” should be “to precise” - done
Lns 299-301- Rephrase the sentence!
(A11) Thanks, we rephrased this sentence:
“So far, there is a significant lack of available scRNAseq data for IIS cells of teleosts, except zebrafish. However, we highlight their high value in the context of IIS evolution and ecological adaptations. These datasets are extensively described in Supplementary Table 3.”
Lns 356-360- modify the sentences!
(A12) Thanks, we modified this sentence:
The main steps of the meta-analysis, including batch corrections and clustering, could be obtained with the functions of multiple-dataset integration implemented in the Seurat [144] and Harmony [145] packages. For annotation of IIS cell types, researchers could use specific packages such as singleR [146] and reference datasets for adult [58] and embryonic stages of zebrafish [115].
Ln 368- “was reviewed” should be “were reviewed” - done
Lns 372-373 - rewrite the sentence!
(A13) Thanks, we rewrited this sentence:
“The reconstruction of regulatory gene networks related to specific IIS cell types and immune processes is an effective and descriptive approach that could be used in this area.”
Lns 380 – 382- Rephrase the sentence!
(A14) Thanks, we rephrased this sentence:“Also, such data could help establish relationships between the ecological niches of teleosts and the peculiarities of their IIS organization.”
Ln 421 – “specie to viruses” should be “species to viruses” - done
Ln 423 – “for study” should be “for studying “ - done
Ln 424- “lifespan” should be “lifespans” - done
Thank you so much again! We corrected these errors and extensively improved overall text quality.
Round 2
Reviewer 1 Report
Comments and Suggestions for Authors
The manuscript has been well improved.
Concerning the title of the manuscript, It focuses on the Teleostei fish. So, please rethink to include the word 'teleostei' or fish in the title.
Author Response
Dear reviewer,
We thank you sincerely for your valuable comments about our work.
According to your suggestion, we decided to modify the title of our manuscript to "Fishing the innate immune system properties through transcriptomic single-cell data of Teleostei". We also did minor revisions to our paper as well as English language corrections.
Reviewer 2 Report
Comments and Suggestions for Authors
The current version has been modified well as reviewers' comments and should be considered to publish.
Author Response
Dear reviewer,
We thank you sincerely for your valuable comments about our work.
According the comments from other reviewer, we decided to modify the title of our manuscript to "Fishing the innate immune system properties through transcriptomic single-cell data of Teleostei". We think this reflects the general message of our review. We also did minor revisions to our paper as well as English language corrections.
Reviewer 3 Report
Comments and Suggestions for Authors
The authors made the requested changes by improving the description of the individual sections (also thanks to the English revisions) and the overall organization of the article. My recommendation is to accept the article in the present form.
Author Response

(The authors gave the same response as above.)

Reviewer 4 Report
Comments and Suggestions for Authors
Comments:
The manuscript (biology-2687880-V2) has been well-improved according to the previous comments and I would recommend it for publication.
Comments on the Quality of English LanguageMinor editing of English language required
Author Response
Dear reviewer,
We thank you sincerely for your valuable comments about our work.
According to the comments from other reviewer, we decided to modify the title of our manuscript to "Fishing the innate immune system properties through transcriptomic single-cell data of Teleostei". We think this reflects the general message of our review. We also did a minor editing of English to make it more understandable for readers.